# Spectra stable blue perovskite light-emitting diodes

Yuanzhi Jiang [1,6], Chaochao Qin[2,6], Minghuan Cui[2], Tingwei He[1], Kaikai Liu[2], Yanmin Huang[1], Menghui Luo[1], Li Zhang[1], Hongyu Xu[1], Saisai Li[1], Junli Wei[1], Zhiyong Liu[2], Huanhua Wang[3], Gi-Hwan Kim[4], Mingjian Yuan[1,5] & Jun Chen[1,5]

Device performance and in particular device stability for blue perovskite light-emitting diodes (PeLEDs) remain considerable challenges for the whole community. In this manuscript, we conceive an approach by tuning the 'A-site' cation composition of perovskites to develop blue-emitters. We herein report a Rubidium-Cesium alloyed, quasi-two-dimensional perovskite and demonstrate its great potential for pure-blue PeLED applications. Composition engineering and in-situ passivation are conducted to further improve the material's emission property and stabilities. Consequently, we get a prominent film photoluminescence quantum yield of around 82% under low excitation density. Encouraged by these findings, we finally achieve a spectra-stable blue PeLED with the peak external quantum efficiency of 1.35% and a half-lifetime of 14.5 min, representing the most efficient and stable pure-blue PeLEDs reported so far. The strategy is also demonstrated to be able to generate efficient perovskite blue emitters and PeLEDs in the whole blue spectral region (from 454 to 492 nm).

[1] Key Laboratory of Advanced Energy Materials Chemistry (Ministry of Education), College of Chemistry, Nankai University, 300071 Tianjin, China. [2] College of Physics and Materials Science, Henan Normal University, 453007 Xinxiang, China. [3] Institute of High Energy Physics, Chinese Academy of Science, 100049 Beijing, China. [4] Photonic Energy Research Center, Korea Photonics Technology Institute, 108, Cheomdan venture-ro, Buk-gu, Gwangju 500-779, Republic of Korea. [5] Renewable Energy Conversion and Storage Center (RECAST), Nankai University, 300071 Tianjin, China. [6] These authors contributed equally: Yuanzhi Jiang, Chaochao Qin. Correspondence and requests for materials should be addressed to M.Y. (email: yuanmj@nankai.edu.cn)

Perovskite light-emitting diodes (PeLEDs) attract significant attention because of their high photoluminescence quantum yields (PLQYs), tunable light emission, and high color purity[1–10]. Extensive efforts have been made on improving the device performance of PeLEDs. Very recently, green-emitting, red-emitting, and near-infrared-emitting PeLEDs have obtained impressive external quantum efficiencies (EQEs) exceeding 20%, which represents an important step toward the commercial application[11–13]. However, blue PeLEDs, as one of the most important prerequisites, tremendously impede the progress in commercialization[14–16]. Device performance, and in particular the device stabilities of blue PeLEDs, remain considerable challenges for the entire PeLED community[17–21].

In general, two approaches have been developed to obtain blue PeLEDs. The first one relies on incorporating chlorine into bromine-based perovskites to tune the bandgap[22–27]. However, these mixed-halide perovskites exhibit intrinsic phase instabilities, especially under electrical potential[20], which is induced by the halide migration. Consequently, the electroluminescence (EL) spontaneously transforms from blue- to green color within minutes under device operation[22–24]. The rapid EL degradation has been observed in almost all of the PeLEDs based on 'mixed-halide' strategy[28,29].

Modulating the quantum-well structure to achieve reduced-dimensional (quasi-2D) perovskites represents another effective approach to generate blue PeLEDs[2,3]. By keeping the 'quantum-well' structures stay in an extremely low thickness region ($n \leq 2$), blue emission can be achieved[20–33]. However, PeLEDs obtained through this strategy commonly exhibit extremely low EQEs (<0.1%), together with broad multiple EL peaks, because of the inefficient internal energy transfer. Moreover, Ostwald ripening typically takes place within minutes that leads to EL spectra significantly red shift[16,32]. Thus, this extremely low $n$ values ($n \leq 2$) quasi-2D perovskites suffer severely from poor spectral stabilities too[32]. Recently, Sargent et al. demonstrate a state-of-the-art sky-blue PeLED following a similar strategy and achieve an EQE of up to 1.5%, together with reasonable spectral stability[18]. However, in their report, the EL spectra cannot reach the real blue region according to the NTSC standard, which impedes their wider applications.

The prior poor device performance and low spectra stabilities for blue PeLEDs are urgently required for a practical remedy. We herein report an approach to engineering 'A-site' cation composition in perovskites to generate blue emitters with good stabilities. We fabricate a Rb–Cs alloyed, $\langle n \rangle_{Rb0.6} = 3$ quasi-2D perovskite and demonstrate its great potential for further pure-blue LED application. Composition engineering and in situ passivation are conducted to further improve its emission property and spectra stabilities. We achieve a prominent film PLQY of around 82% at 476 nm, with excitation density as low as 1.5 mW cm$^{-2}$. Moreover, no PL spectra shift or broadening have been observed in these materials even after long-time thermal annealing, which demonstrates excellent material stabilities. Eventually, PeLEDs that are displaying pure-blue emission at 475 nm with a narrow full width at half maximum (FWHM) of around 20 nm, peak brightness of 100.6 cd m$^{-2}$, and peak EQE of 1.35% are accomplished. Half-lifetime ($T_{50}$) measurements at the maximum EQE points reveal a lifetime of 14.5 min. These values represent a state-of-the-art EQE and the highest stability among the blue PeLEDs reported so far[16,18,22–27,30,31,33–35]. Finally, based on this 'A-site' cation composition engineering, we also demonstrate the full spectrum emission perovskites in the whole blue region (from 454 to 492 nm) to meet the various requirements from different industries.

## Results

**Properties of 3D Rb–Cs alloyed perovskites.** Rubidium (Rb$^+$, 152 pm) possesses a smaller ionic radius compared with cesium (Cs$^+$, 167 pm)[36]. Thus, incorporating Rb$^+$ into 3D CsPbBr$_3$ perovskites is expected to afford alloyed perovskites, Rb$_x$Cs$_{1-x}$PbBr$_3$, which possess a larger bandgap due to the octahedral titling. Rb$_x$Cs$_{1-x}$PbBr$_3$ ($0 \leq x \leq 1$) perovskites (Supplementary Fig. 1a) are then fabricated through a solution-based process. As expected, the optical bandgap of Rb$_x$Cs$_{1-x}$PbBr$_3$ films does increase from 2.31 to 2.60 eV ($0 \leq x \leq 0.8$) with an increasing amount of Rb$^+$ (Supplementary Figs. 2a, 29)[37]. However, the stability of Rb$_x$Cs$_{1-x}$PbBr$_3$ declines as the Rb$^+$ ratio increases, which ascribes to the increased octahedral distortion caused by continuous Rb$^+$ substitution (Supplementary Fig. 2b)[37]. Accordingly, Rb$_{0.6}$Cs$_{0.4}$PbBr$_3$ is the only stabilized blue emitter among the whole series (Fig. 1c, Supplementary Fig. 3). Unfortunately, Rb$_{0.6}$Cs$_{0.4}$PbBr$_3$ exhibits a low PLQY of 0.7% with a fast PL lifetime of around 0.4 ns, illustrating that a strong trap-mediated non-radiative recombination takes place (Supplementary Fig. 4)[10]. Furthermore, the rough and uneven film morphology is also far from demand (Supplementary Fig. 5).

**Quasi-2D structure for Rb–Cs alloyed perovskites.** Anyhow, the low PLQY and PL wavelength (499 nm) diverging from an ideal pure-blue spectral region (from 460 to 480 nm) require a solution. As documented, quasi-2D perovskites exhibit much-improved PLQYs and blue-shifted PL emission compared with their 3D analogs, owing to the efficient energy transfer[2,3]. Bearing this in mind, we conceive to fabricate $\langle n \rangle = 3$ PEA$_2$(Rb$_x$Cs$_{1-x}$)$_2$Pb$_3$Br$_{10}$ ($0 \leq x \leq 1$) perovskites with a different Rb$^+$ content in order o obtain efficient blue perovskite emitters (Fig. 1a and Supplementary Fig. 1b). Multiple excitonic absorption peaks are observed in the absorption spectra of $\langle n \rangle = 3$ PEA$_2$(Rb$_x$Cs$_{1-x}$)$_2$Pb$_3$Br$_{10}$ ($0 \leq x \leq 1$), corresponding to the perovskite species with different $n$ values (Fig. 1b) (note: we use '$\langle n \rangle$' that represents a quasi-2D perovskite domain but '$n$' stands for the species with a fixed composition). These characteristics are consistent with previous reports, confirming that quasi-2D perovskites contain a variety of $n$ species rather than a single component[2,38]. Meanwhile, XRD patterns exhibit different diffraction peaks compared with their 3D analogs caused by the layered structure (Supplementary Figs. 6, 7c, 8 and Supplementary Table 2)[39].

As expected, PL spectra for pristine $\langle n \rangle = 3$ perovskites are shown in Fig. 1c, which do exhibit a significant blue shift compared with their 3D counterparts (Supplementary Fig. 29)[2,40]. For instance, the pristine $\langle n \rangle = 3$ PEA$_2$(Rb$_{0.6}$Cs$_{0.4}$)$_2$Pb$_3$Br$_{10}$ (denoted as $\langle n \rangle_{Rb0.6} = 3$) perovskite displays a single PL peak at 466 nm with a FWHM of around 25 nm. In particular, significant PL enhancement (more than 10-fold) is observed for the pristine $\langle n \rangle_{Rb0.6} = 3$ perovskite when compared with its 3D analog (Fig. 1c), and herein, a PLQY of 11% is obtained with excitation density as low as 1.5 mW cm$^{-2}$. Meanwhile, we also systematically investigate the emission property of the quasi-2D PEA$_2$(Rb$_{0.6}$Cs$_{0.4}$)$_{n-1}$Pb$_n$Br$_{3n+1}$ perovskite with the other $\langle n \rangle$ values, and confirm that it is not suitable for further blue PeLED application (Supplementary Figs. 7, 9, 10 and Supplementary Notes 1, 2). Even though the PLQY for pristine $\langle n \rangle_{Rb0.6} = 3$ reaches 11%, it is still far inferior to meet the requirement for PeLED application; thus, further optimization needs to be conducted.

**Optical properties of RbBr-rich quasi-2D perovskites.** It is documented that increased molar proportion of methylammonium bromide (MABr) in MAPbBr$_3$ could prevent the formation of metallic lead, and further passivate the grain boundary, leading to greatly improved PLQYs[1,32]. Taking this into consideration, we

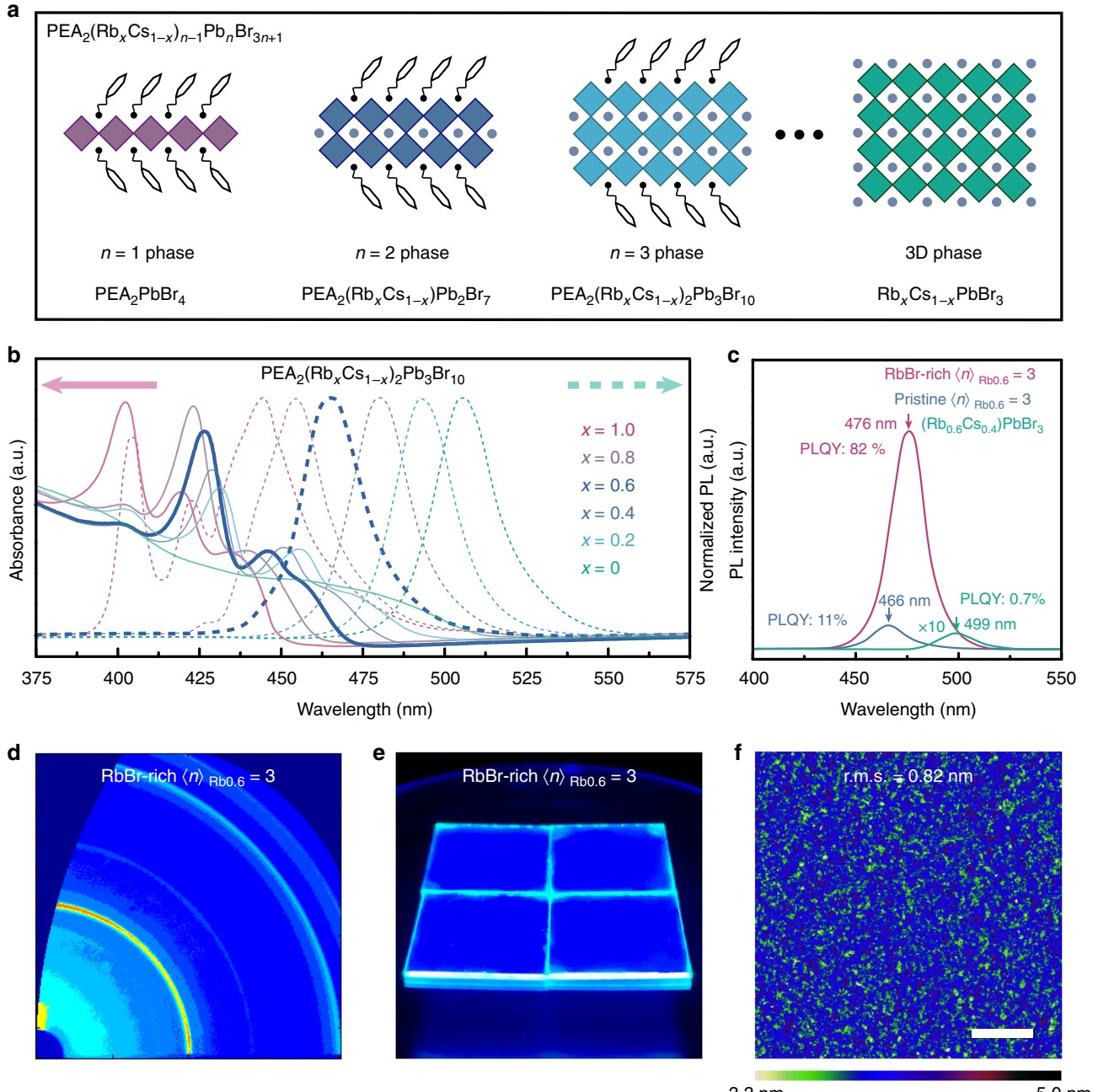

**Fig. 1** Characteristics of perovskite films. **a** Cartoon structure of Cs–Rb alloyed quasi-2D $PEA_2(Rb_xCs_{1-x})_{n-1}Pb_nBr_{3n+1}$ perovskites. **b** UV–vis absorption and steady-state PL spectra of pristine $\langle n \rangle = 3$ $PEA_2(Rb_xCs_{1-x})_2Pb_3Br_{10}$ ($0 \leq x \leq 1$) perovskites. **c** Steady-state PL spectra and PLQYs of 3D perovskite $Rb_{0.6}Cs_{0.4}PbBr_3$, pristine, and RbBr-rich $\langle n \rangle_{Rb0.6} = 3$ films. **d** GIWAXS pattern of the RbBr-rich $\langle n \rangle_{Rb0.6} = 3$ film. **e** Photoluminescence image of RbBr-rich $\langle n \rangle_{Rb0.6} = 3$ films under excitation. **f** AFM height image of RbBr-rich $\langle n \rangle_{Rb0.6} = 3$ film with a scale bar of 2 μm

thus conceive to incorporate an excess amount of RbBr into the pristine $\langle n \rangle_{Rb0.6} = 3$ perovskite to form the RbBr-rich $\langle n \rangle_{Rb0.6} = 3$ perovskite and then investigate its emission property. Meanwhile, excessive CsBr could essentially induce a cation exchange, leading to a green emitter, which is not interested in this research (Supplementary Fig. 11). As shown, XRD patterns confirm that the RbBr-rich $\langle n \rangle_{Rb0.6} = 3$ perovskite possesses a very similar diffraction pattern as the pristine one (Supplementary Fig. 12), and no extra RbBr peaks can be detected (Supplementary Fig. 13). Strikingly, we find that the PLQY of RbBr-rich $\langle n \rangle_{Rb0.6} = 3$ perovskite soars up to 82% at the same excitation density (1.5 mW $cm^{-2}$) with an excellent stability (Supplementary Figs. 14, 15, 28, and 30). Moreover, a 10-nm red shift in PL spectra compared

with the pristine $\langle n \rangle_{Rb0.6} = 3$ perovskite is observed (Fig. 1c, Supplementary Figs. 16d, 28). Notably, an immoderate amount of excess RbBr should be avoided, or it could induce a serious defect state, as well as an irreversible morphology destruction (Supplementary Figs. 17, 18).

**Photocarrier transportation and recombination dynamics.** Transient absorption (TA) measurement is conducted to investigate the transfer and recombination dynamics of photogenerated carriers in pristine and RbBr-rich $\langle n \rangle_{Rb0.6} = 3$ quasi-2D perovskites. As shown in Fig. 2a and b, three distinctive ground-state bleach (GSB) peaks are observed in the pristine $\langle n \rangle_{Rb0.6} = 3$ perovskite. TA spectra at different selected delay times are

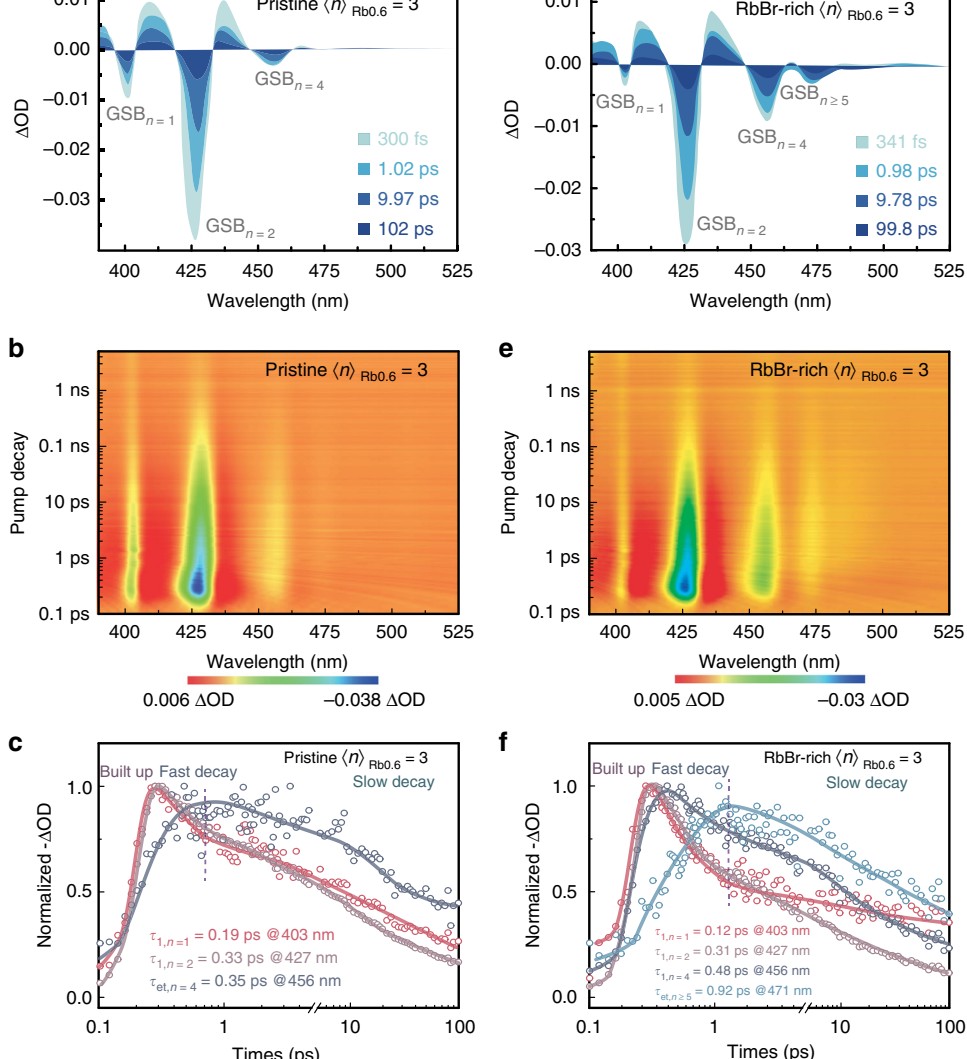

**Fig. 2** TA measurements for quasi-2D perovskite films. **a**, **d** TA spectra at selected timescales; **b**, **e** time–wavelength-dependent TA color maps; and **c**, **f** TA spectra at a different wavelength as a function of delay time for pristine and RbBr-rich $\langle n \rangle_{Rb0.6} = 3$ films, respectively

displayed in Fig. 2a. It is clear that photogenerated excitons are initially formed at $n = 1$ and $n = 2$ species, and the $GSB_n = 4$ gradually emerges with increasing the decay time, indicating that an energy transfer process takes place[2,6,41]. Notably, the GSB of $n = 3$ species could not be obviously distinguished, probably due to the closely packed GSB peaks and strong excited state absorption (ESA) signals (Supplementary Figs. 19, 20 and Supplementary Note 3). The decay kinetics of each GSB is extracted in Fig. 2c. The fast decay components for $n = 1$ and 2 bleaching are ascribed to the carrier transfer from lower-$n$ species (donor domain) to the emitting domain ($n = 4$ species); the slow decay component shall refer to the charge trapping according to previous reports[42]. The fast decay component reveals time constants ($\tau_1$) of 0.19 ($n = 1$) and 0.33 ps ($n = 2$), respectively. Coincidentally, the fitted time constant ($\tau_{et}$) of the rising component for $GSB_n = 4$ is 0.35 ps, which is in good agreement with a fast decay timescale. Accordingly, we can conclude that carrier transfer from donor domains ($n = 1$ and 2) to the emitting domain ($n = 4$) is accomplished in picoseconds with near-unit efficiency.

TA characterization for the RbBr-rich $\langle n \rangle_{Rb0.6} = 3$ perovskite is shown in Fig. 2d, e. Notably, a new GSB peak domain at 471 nm referring to $n = 5$ and $n > 5$ species gradually appears, which

cannot be observed in the pristine $\langle n \rangle_{Rb0.6} = 3$ perovskite. The TA data suggest that new perovskite domains, $n = 5$ and $n > 5$ species, appear when introducing an excess of RbBr into pristine $\langle n \rangle_{Rb0.6} = 3$. The decay kinetics for different GSB peaks correlate well with the characteristics of energy transfer (Fig. 2f and Supplementary Table 3). In addition, the population in $n = 1$, 2, and 4 species decays faster than carrier injection speed, and the decay kinetics also exhibit a slower lifetime for the lower bandgap species. The GSB maximum sequentially delays for the lower bandgap species ($n \geq 5$), proving that lower bandgap domains have received carriers from the larger bandgap species. All these details together confirm that carriers have transferred from a wider to a lower bandgap species, and eventually recombined in the $n > 5$ domain[42,43]. This more-graded energy landscape leads to a more efficient energy transfer compared with the case of pristine $\langle n \rangle_{Rb0.6} = 3$ that facilitates the radiative recombination.

Time-resolved PL (TRPL) analysis is in excellent agreement with the TA kinetics (Fig. 3a, b). The PL maximum of pristine $\langle n \rangle_{Rb0.6} = 3$ perovskite moves from 466 to 476 nm after the RbBr-rich $\langle n \rangle_{Rb0.6} = 3$ perovskite formed. We cannot extract the PL lifetime for each higher-bandgap species ($n = 1$, 2, and 4) to illustrate energy transfer kinetics because of the instrument limitation. The overall PL lifetime for RbBr-rich $\langle n \rangle_{Rb0.6} = 3$

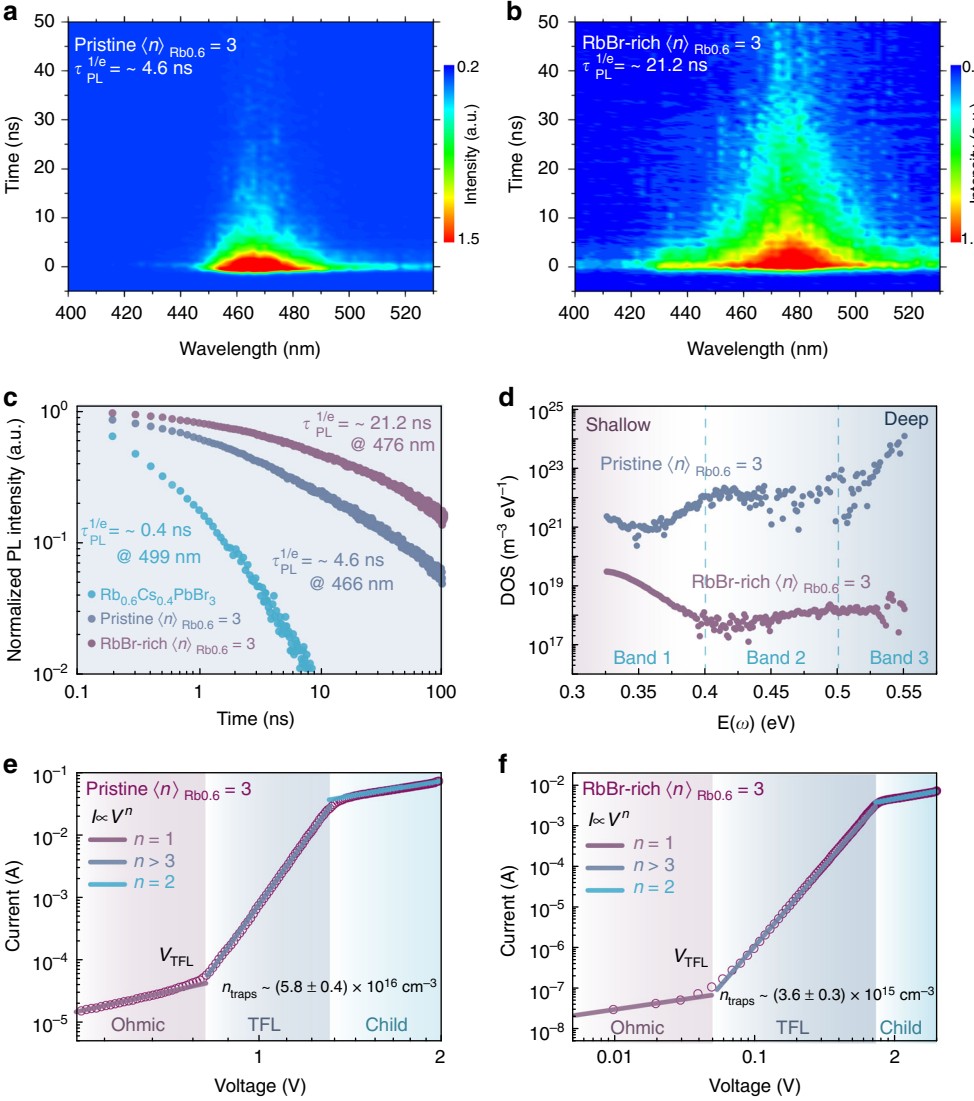

**Fig. 3** Characteristics of trap densities. TPRL measurements of (**a**) pristine and (**b**) RbBr-rich $\langle n \rangle_{Rb0.6} = 3$ films. **c** The PL lifetime for 3D $Rb_{0.6}Cs_{0.4}PbBr_3$, pristine, and RbBr-rich $\langle n \rangle_{Rb0.6} = 3$ films. **d** Trap density of states (*t*DOS) for pristine and RbBr-rich $\langle n \rangle_{Rb0.6} = 3$ perovskite in TAS analysis. Current–voltage (*J*–*V*) response of (**e**) pristine and (**f**) RbBr-rich $\langle n \rangle_{Rb0.6} = 3$ films in SCLC measurement

perovskite was determined to be 21.2 ns, which is around five times as long as the pristine $\langle n \rangle_{Rb0.6} = 3$ perovskite's lifetime (4.6 ns) (Fig. 3c). The longer PL lifetime generally indicates a lower defect density, which should account for the primary reason for PLQY enhancement[5].

**Electrical characterization of quasi-2D perovskites.** Herein, space-charge-limited current (SCLC) and thermal admittance spectroscopy (TAS) technique are used to quantize the defect density. SCLC measurements are conducted with the device structure of ITO/perovskites/MoO₃/Au. We then measure the *J*–*V* response of the different perovskite films in the SCLC regime (Fig. 3e, f). Three distinct regions clearly identify from the *J*–*V* curves. At intermediate voltages, the current injection displays a marked nonlinear increase (set in at $V_{TFL} = 0.82$ V for the pristine one, and 0.05 V for the RbBr-rich one), and curves then switch to the trap-filled-limit (TFL) region, in which all the trap density levels are continuously filled by the increasing voltage. Thus, the onset voltage ($V_{TFL}$) is proportional to the density of trap states ($n_{traps}$). Consequently, we extract a trap state density of

$n_{traps} = 3.6 \times 10^{15}$ cm$^{-3}$ for RbBr-rich $\langle n \rangle_{Rb0.6} = 3$ perovskite, which is more than one order of magnitude lower than the pristine $\langle n \rangle_{Rb0.6} = 3$ perovskite ($n_{traps} = 5.8 \times 10^{16}$ cm$^{-3}$)[44].

Furthermore, we also extract the trap density from TAS analysis. TAS technique is recognized as an effective method for characterizing both shallow and deep defects. In general, the energetic profile of trap density of states (*t*DOS) can be derived from the angular frequency-dependent capacitance, where the energetic demarcation $E(\omega)$ is correlated with the trap depth. As revealed in Fig. 3d, the RbBr-rich $\langle n \rangle n_{Rb0.6} = 3$ perovskite exhibits a much lower *t*DOS than the pristine one over the whole trap depth. In particular, the RbBr-rich $\langle n \rangle_{Rb0.6} = 3$ perovskite displays around four orders of magnitude lower deeper trap density (from 0.50 to 0.55 V) than its pristine analog. The marked decreased *t*DOS in TAS analysis is consistent with the SCLC data, indicating that RbBr effectively passivates the defect states, especially deep trap states in the pristine $\langle n \rangle_{Rb0.6} = 3$ perovskite.

Based on the above optical and electrical analysis, we thus conclude that the excessive amount of RbBr actually plays dual roles in improving the pristine $\langle n \rangle_{Rb0.6} = 3$ perovskite's

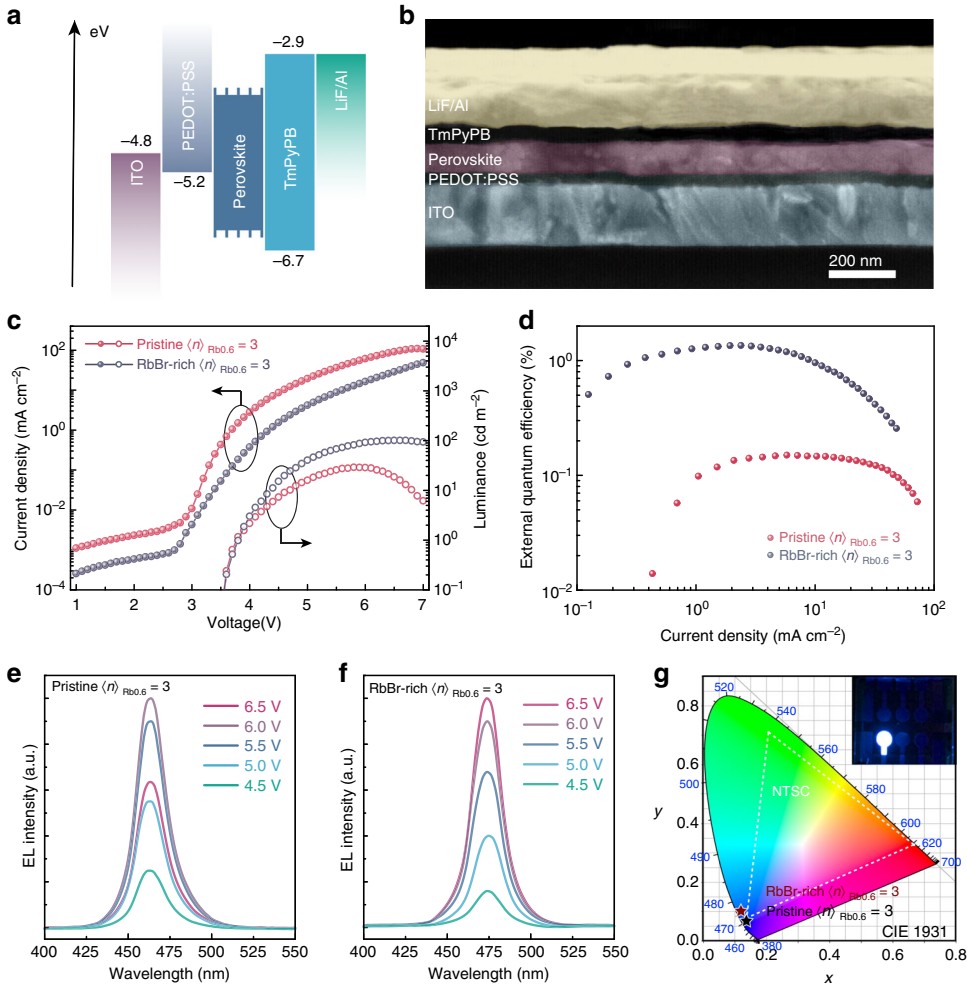

**Fig. 4** Optoelectronic characteristics of the PeLEDs. **a** Band alignment of each function layer in devices. **b** SEM cross-sectional image of the RbBr-rich $\langle n \rangle_{Rb0.6} = 3$ based devices. **c** $I$–$V$ and $L$–$V$ curves for pristine and RbBr-rich $\langle n \rangle_{Rb0.6} = 3$ devices. **d** EQE characteristics of pristine and RbBr-rich $\langle n \rangle_{Rb0.6} = 3$ devices. EL spectra of (**e**) pristine and (**f**) RbBr-rich $\langle n \rangle_{Rb0.6} = 3$ at different voltage bias. **g** The corresponding Commission Internationale de l'Echlaiage (CIE) coordinate of pristine and RbBr-rich $\langle n \rangle_{Rb0.6} = 3$ devices, and the inset image shows the electroluminescence of RbBr-rich $<n>_{Rb0.6} = 3$ based PeLEDs

emission property. First, it induces the $n$-value shift to the larger number and consequently forms new $n = 5$ and $n > 5$ species. The phenomena are in consistent with previous reports, wherein incorporating an excess of FAI cation in quasi-2D $NMA_2FA_{n-1}Pb_nI_{3n+1}$ perovskites can induce the $n$ values to slightly shift to the larger number[6]. A more-graded energy landscape is herein formed in the RbBr-rich $\langle n \rangle_{Rb0.6} = 3$ perovskite[41]. Consequently, funneling carriers into the low concentration $n > 5$ subpopulation could facilitate radiative recombination, leading to PLQY enhancement, together with a slightly spectral redshift (around 10 nm) (Supplementary Fig. 21). On the other hand, RbBr effectively passivates the pristine $\langle n \rangle_{Rb0.6} = 3$'s defect states, leading to overall several orders of magnitude lower defect density. The dramatically reduced defect density accounts for the primary reason of the PLQY enhancement[5]. Ultimately, the RbBr-rich $\langle n \rangle_{Rb0.6} = 3$ perovskite film exhibits a prominent PLQY of around 82% at 476 nm, with low excitation power density of 1.5 mW cm$^{-2}$.

**Device structure and performance**. Encouraged by the above findings, we further fabricated and optimized the PeLED device using different perovskite films. We configured PeLEDs as the structure glass/ITO/PEDOT:PSS (30 nm)/perovskites (80 nm)/

TmPyPB (40 nm)/LiF (0.8 nm)/Al (100 nm). The schematic of the band alignment diagram for all function layers is shown in Fig. 4a. The band structure of RbBr-rich $\langle n \rangle_{Rb0.6} = 3$ film was determined using ultraviolet photoelectron spectroscopy (UPS; Supplementary Fig. 22). The cross-section scanning electron microscopy (SEM) image of the completed device also confirms the stacked film thickness (Fig. 4b).

Compared with the pristine $<n>_{Rb0.6} = 3$ perovskite, an extremely dense and uniform film is observed in SEM images for the RbBr-rich $<n>_{Rb0.6} = 3$ perovskite (Supplementary Fig. 23). As shown in the atomic force microscopy (AFM) image, the surface roughness of the RbBr-rich $<n>_{Rb0.6} = 3$ perovskite film on PEDOT:PSS is only 0.82 nm (Fig. 1f, Supplementary Fig. 24), which is sufficiently smooth to avoid current leakage. In addition, grazing-incidence wide-angle X-ray scattering (GIWAXS) measurements are conducted. As revealed in Fig. 1d, uniform intensities along the Debye–Scherrer rings are observed, indicating that the quasi-2D perovskite crystallites are isotropically oriented in the film[13].

Pristine $<n>_{Rb0.6} = 3$ devices exhibit low performance with 0.15% EQE and 28.9 cd m$^{-2}$ luminance (Fig. 4d), which is not a surprise in view of its low PLQY. In addition, the device displays a brightness rolloff at around 6 V (Fig. 4c), indicating that a strong non-radiative recombination takes place. Strikingly, by taking

advantage of high PLQY and advanced film quality, the RbBr-rich $\langle n \rangle_{Rb0.6} = 3$ perovskite displays a significant enhancement in EQE of 1.35%, and luminance of 100.6 cd m$^{-2}$ (Fig. 4d, Supplementary Fig. 25) with good reproducibility (Supplementary Fig. 26). The resulting EL spectra closely match its PL spectra, preserving a narrowband emission. The narrowband emission exhibits excellent color purity with CIE chromatic coordinates of (0.115, 0.099) close to NTSC standard values (Fig. 4g), and follows a Lambertian profile (Supplementary Fig. 27). Notably, the result represents the most efficient pure-blue PeLED reported so far, which is one-order-of-magnitude enhancement in EQE over the best-reported pure-blue PeLEDs operating in the similar wavelength.

As far as we know, to achieve excellent spectra stability for blue PeLEDs remains a global challenge, owing to the higher driving voltage required. Spectral stability for our pristine and RbBr-rich $\langle n \rangle_{Rb0.6} = 3$ PeLEDs is recorded at different bias voltage. The EL spectra as a function of bias voltage are shown in Fig. 4e, f. As bias voltage increased from 4.5 to 6 V, the EL intensity increased correspondingly, and the EL peak position and FWHM remained unchanged. When the bias voltage further increased up to 6.5 V, the EL intensity of RbBr-rich $\langle n \rangle_{Rb0.6} = 3$ still increased and exhibited excellent spectral stability. On the other hand, the spectra of pristine $\langle n \rangle_{Rb0.6} = 3$ also remained unchanged at 6.5 V, in spite of the brightness rolloff. These results undoubtedly demonstrate the excellent spectral stability of the quasi-2D Rb–Cs alloyed perovskites, which have overcome the previous instabilities induced by the halide migration or Ostwald ripening.

**Spectral stability of quasi-2D Rb–Cs alloyed perovskites.** In order to further demonstrate the spectral stability of the perovskites, the pristine and RbBr-rich $\langle n \rangle_{Rb0.6} = 3$ perovskite films are first tested under thermal annealing. As shown in Fig. 5a, the

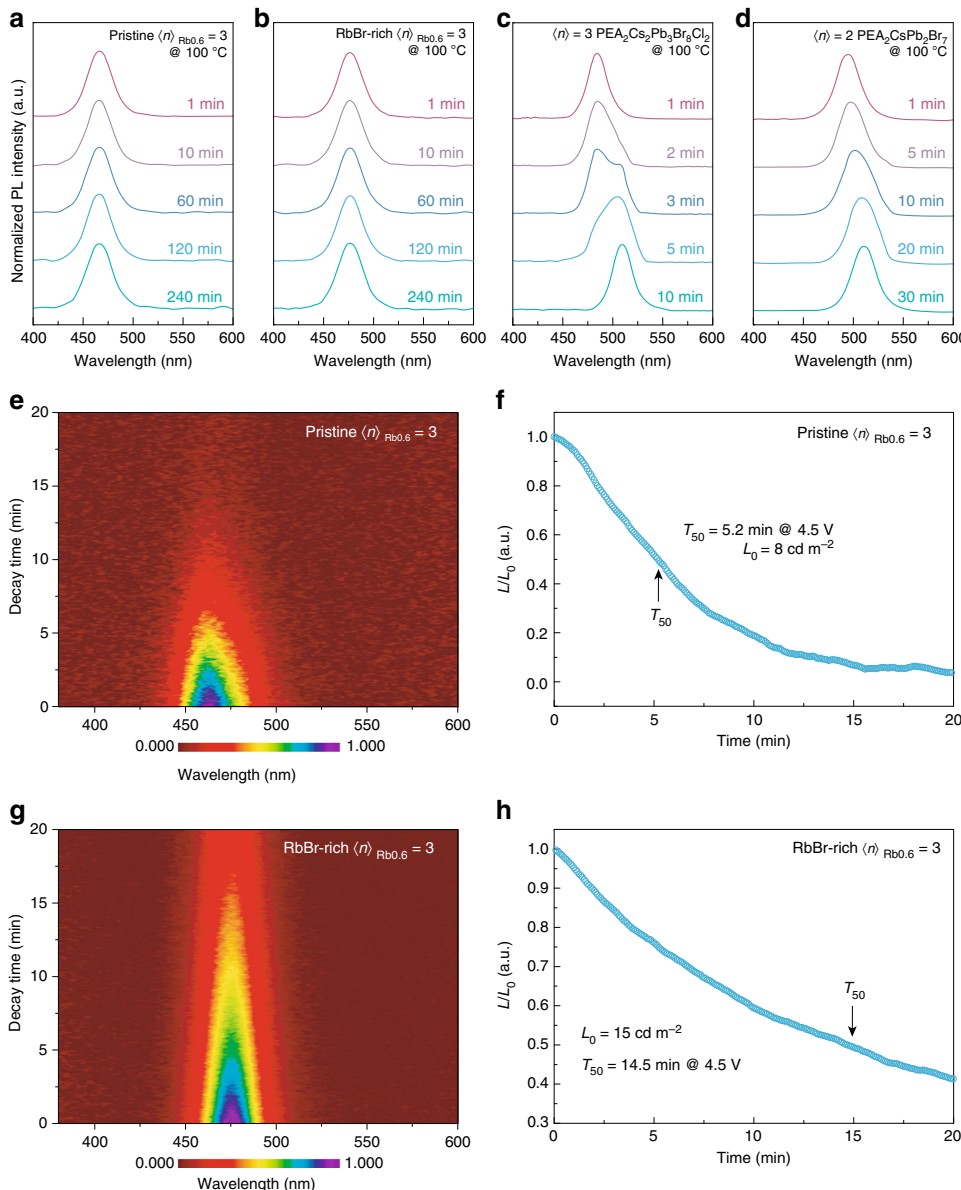

**Fig. 5** Spectral stability measurements. PL spectra evolution of (**a**) pristine $\langle n \rangle_{Rb0.6} = 3$, (**b**) RbBr-rich $\langle n \rangle_{Rb0.6} = 3$, (**c**) $\langle n \rangle = 3$ mixed-halide PEA$_2$Cs$_2$Pb$_3$Br$_8$Cl$_2$, and (**d**) lower-dimensional $\langle n \rangle = 2$ PEA$_2$CsPb$_2$Br$_7$ perovskites after continuous thermal treatment (100 ℃) for different times. **e, g** The EL spectra change under constant voltage of 4.5 V. Lifetime measurement ($T_{50}$) of the (**f**) pristine and (**h**) RbBr-rich $\langle n \rangle_{Rb0.6} = 3$ PeLED device at a constant voltage of 4.5 V

PL spectra of the pristine $<n>_{Rb0.6} = 3$ perovskite remained unchanged after heating at 100 °C for 4 h. Similarly, the RbBr-rich $<n>_{Rb0.6} = 3$ perovskite illustrates excellent spectral stability too, in which both the PL peak position and FWHM changed negligibly after 4 h of annealing (Fig. 5b). In contrast to the previously reported, mixed-halide blue emitter, $PEA_2Cs_2Pb_3(Br_{0.8}Cl_{0.2})_{10}$ exhibited poor spectral stability, in which a continuous spectra redshift was observed minutes after annealing began (Fig. 5c). The same phenomena were detected for $<n> = 2$ quasi-2D perovskite $PEA_2CsPb_2Br_7$ (Fig. 5d), which were in good agreement with the previous reports, indicating that Ostwald ripening probably occurred, which is caused by the Joule heating[35].

Furthermore, we record the spectral stability of the PeLEDs under continuous operation at a constant voltage of 4.5 V to extract its lifetime. The half-lifetime ($T_{50}$) was defined as the time required for the luminance to decease to 50% of the start luminance ($L_0$)[11]. In brief, the EL spectra changes negligibly under continuous operation for 20 min (Fig. 5e, g)[22–24,27]. The $T_{50}$ for pristine and RbBr-rich $<n>_{Rb0.6} = 3$ PeLEDs are determined to be 5.2 and 14.5 min, respectively (Fig. 5f, h). The result represents the most stable blue PeLEDs reported so far (Supplementary Table 4), which is comparable with the reported stability of green and red analogs[6,8,20,45–50].

## Discussion

As mentioned above, the ratio of Rb to Cs, thickness of the quantum wells ($<n>$ values), as well as content of excess RbBr synergistically determined the optical properties of the resulting perovskite, including emission wavelength, PLQYs, and spectral stability. Keeping these in mind, by tuning the variable composition, we are able to achieve highly emissive perovskite emitters in the entire blue spectral region (from 454 to 492 nm) (Supplementary Fig. 28 and Supplementary Note 4). In particular, the $PEA_2(Rb_{0.8}Cs_{0.2})Pb_3Br_{10}$ (RbBr-rich $<n>_{Rb0.8} = 3$) and $PEA_2(Rb_{0.4}Cs_{0.6})Pb_3Br_{10}$ (RbBr-rich $<n>_{Rb0.4} = 3$) exhibit PLQYs of 58.3% and 84.1% after RbBr treatment, with an emission at 458 nm (deep blue) and 490 nm (sky blue), respectively (Supplementary Figs. 20, 28). Meanwhile, we tracked PLQY for RbBr-rich $<n>_{Rb0.8} = 3$ and RbBr-rich $<n>_{Rb0.4} = 3$ under ambient condition. As expected, the materials' stability significantly reduced when the ratio of Rb to Cs was increased (Supplementary Fig. 30), which is consistent with our previous hypothesis. Furthermore, we also fabricated a spectra-stable sky-blue PeLED through the same technique to demonstrate its versatile application. The sky-blue PeLEDs exhibited a peak EQE of 1.48%, peak luminance of 854.3 cd m$^{-2}$, and $T_{50}$ lifetime of 18.7 min without systematic optimization, representing one of the most stable sky-blue PeLEDs (Supplementary Figs. 31, 32, Supplementary Table 4, and Supplementary Note 5).

In summary, we developed an approach to obtain spectra-stable blue-emitting perovskite films with a relatively high PLQY. The approach focuses on engineering the 'A-site' cation to obtain blue emission, which is confirmed to be able to generate stable blue perovskite emitters. We herein designed a Rb–Cs alloyed, quasi-2D $<n>_{Rb0.6} = 3$ perovskite and demonstrated its potential for pure-blue LED application. Composition engineering and in situ passivation are then conducted to regulate its emission property. We achieved a prominent film PLQY of around 82% for our RbBr-rich $<n>_{Rb0.6} = 3$ perovskite under low excitation density. Encouraged by these findings, we finally achieved a spectra-stable pure-blue PeLED with the peak EQE of 1.35% and a lifetime of 14.5 min. These materials have overcome the previous problems suffered by the whole community, representing the most efficient and stable blue PeLEDs reported so far. This concept is also demonstrated to be useful in the whole blue emission region and expected to open an avenue for the PeLED community.

## Methods

**Materials.** PEDOT:PSS (Clevios P VP AL 4083) was purchased from Heraeus. RbBr (99.6%), CsBr (99.999%), and PbBr$_2$ (99.999%) were bought from Sigma-Aldrich. PEABr (99.999%) was purchased from Greatcell Solar. TmPyPB (99%) and LiF (99.99%) were purchased from Lumtech Corp. All of the reagents were directly used without any purification.

**LED device fabrication.** The precursor solutions were filtered by 0.45-μm poly (tetrafluoroethylene) filters before using. ITO substrates (10 Ω sq$^{-1}$) were cleaned by standard procedures and then treated with O$_2$ plasma for 15 min. PEDOT:PSS was then spin-coated on the substrate at 5000 r.p.m. for 30 s, followed by annealing at 150 °C for 20 min. Then, the substrates were transferred to a nitrogen-filled glovebox. Different DMSO precursor solutions were coated on top of the PEDOT:PSS film via a two-step spin-coating process at 1000 and 8000 r.p.m. for 10 and 30 s, respectively. In the course of the second step, 200 μL of chloroform was dropped onto the film, followed by annealing at 100 °C for another 5 min. Finally, TmPyPB (40 nm), LiF (0.8 nm), and Al electrode (100 nm) were contacted by thermal evaporation. Perovskite precursor solutions were prepared according to Supplementary Table 1.

**Device measurements and film characterization.** All the PeLEDs were measured in a glovebox. A Keithley 2400 source meter coupled with a fiber integration sphere (FOIS-1-FL, QE65 Pro spectrometer) were used. PeLEDs were measured with a scanning rate of 0.1 V s$^{-1}$ and a dwell time of 1 s. The device active area was determined to be 8.57 mm$^2$. The whole measurement method is consistent with the previous reports[12,51]. The performance of PeLEDs was double-checked using a spectroradiometer of PR-750 to ensure the accuracy and precision of the measurements. The device stability tests were conducted in a glovebox. The EL spectra were collected every 5 s. UV–vis adsorption was recorded from LAMBDA 950 UV/Vis/NIR spectrophotometer. XRD patterns were obtained using Bruker D8 diffractometer with Cu Kα radiation ($\lambda = 1.5406$ Å). The morphologies of perovskite films were recorded by field-emission SEM (JSM-7500F, JEOL). AFM measurements were collected in noncontact mode (Dimension Icon, Bruker). GIWAXS pattern was measured on beamline 1W2A at Beijing Synchrotron Radiation Facility (BSRF), China. A monochromatic beam of $\lambda = 1.54$ Å was used, and the incident angle was 0.15°. The fs-TA measurements were carried out on a Helios pump-probe system (Ultrafast Systems LLC) coupled with an amplified femtosecond laser system (Coherent, 35 fs, 1 kHz, 800 nm). The probe pulses (from 380 to 600 nm) were generated by focusing a small portion (around 10 μJ) of the fundamental 800-nm laser pulses into a 1-mm CaF$_2$. The 365-nm pump pulses were generated from an optical parametric amplifier (TOPAS-800-fs). The steady-state PL of the perovskite films was measured by a fluorescence spectrophotometer (Fluoromax 4, Horiba) with an excitation wavelength of 365 nm provided by a 450-W xenon lamp. The TRPL decay lifetimes were acquired via a monochromator/spectrograph (Omni-λ300, Zolix) and an oscilloscope (GDS-3354, GWINSTEK). The 365-nm pump pulses that are used in the TRPL experiments were the same as those used in the broadband femtosecond TA measurements. The PLQY was measured by a Quanta-Phi integrating sphere, which was an optical fiber bundle coupled with a Fluorolog system, and the excitation wavelength was 365 nm, with an excitation intensity of 1.5 mW cm$^{-2}$. The UPS spectra were obtained on Thermo Scientific ESCALab 250Xi.

## Data availability

The data that support the findings of this study are available from the corresponding author upon reasonable request.

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

## Acknowledgements

The authors acknowledge financial support from the National Natural Science Foundation of China (No. 21771114, 11804084), MOE 111 (B12015), the Natural Science Foundation of Tianjin (17JCYBJC40900, 18YFZCGX00580), the Advanced Technology Research Program of Henan Province (182102210369), and the Fundamental Research Funds for the Central Universities. M.Y. acknowledges financial support from Thousand Youth Talents Plan of China.

## Author contributions

M.Y. conceived the idea. J.C. and M.Y. supervised the work. M.Y., G.K., Y.J., Y.H., M.L., L.Z., and J.W. fabricated devices and analyzed the data. C.Q. and M.C. performed TA and TRPL measurements and analyzed the data. T.H., K.L., S.L., and Z.L. performed SEM measurements. H.W. and H.X. performed GIWAXS measurements. Y.J. and M.Y. co-wrote the paper. All authors read and commented on the paper.

## Additional information

**Competing interests:** The authors declare no competing interests.

