## [Peer Review File · Nature Communications]

Reviewers' comments:

Reviewer #1 (Remarks to the Author):

In this manuscript, the authors reported the synthesis of Rb-Cs alloyed perovskite quantum wells for pure-blue LED. The resulted perovskite film and LED device exhibited improved performance in terms of spectral stability, which is a well-known challenge for blue perovskite LED devices. They provide a new concept to construct blue perovskite emitter. This work will be interesting to perovskite community. Thus, the reviewer suggests publishing in Nature Communications after a major revision. More detailed comments are as follows.

1. The authors claimed the addition of excess amount of RbBr could significantly increase the PLQY, and the 100% excess of RbBr lead to the highest PLQY. It seems the PLQY start to decline the mount of RbBr more than 100%. Could the author provide the explanation?
2. Why a 10 nm red-shift in PL spectra of extra RbBr sample compared to the pristine $\text{Rb}_{0.6}\text{Cs}_{0.4}\text{PbBr}_3$ perovskite sample? Did the extra RbBr changed the ratio of Rb : Cs in the perovskite crystal?
3. The authors studied the stability of perovskite film after 10 day (SI Figure 10). How about the PLQY of the perovskite film after 10 days.
4. In the transient absorption (TA) measurements, the authors claimed that the $n \gg 3$ perovskite domains could be produced in pristine $n = 3$ perovskite, and possess higher defect density which is similar with 3D perovskite. Thus, the measurement on 3D $\text{Rb}_{0.6}\text{Cs}_{0.4}\text{PbBr}_3$ perovskite should be also provided to confirm the hypothesis.
5. In steady state absorption, there are four excitonic absorption peaks $n = 1, 2, 3, 4$, why did the TA feature of $n = 3$ phase absent?
6. In order to make the device performance data more convincing, the authors need to provide statistical data.
7. Although the lifetime of perovskite LED was improved, 14.5 min is still in the same region as other reports.
8. the author should obtain some help in polishing the language, there are many grammatical errors, or inappropriate statement. Particularly, there are many exaggerated adjectives through the manuscript, such as superfast, terrible, etc., and missing "that" or "which" for clauses.

Reviewer #2 (Remarks to the Author):

In the manuscript, Jiang et al. report the fabrication of spectra stable and efficient blue perovskite light-emitting diodes. After introducing the Rb into the precursor system, the reduced-dimensional Rb-Cs alloyed perovskite is formed. Quasi-2D structure was proved to be able to stabilize the metastable perovskite phase and facilitate the energy transfer. Detailed energy transfer/combination kinetics was carefully investigated through transient spectroscopy. After in-situ passivation, a high EQE of 1.35% with excellent spectra stability was obtained. Overall, this work provides a new strategy to generate spectra stable blue perovskite LED. The paper is interesting and will be helpful to researchers in related fields. Therefore, I recommend the paper published in Nature Communications after minor revision as follows:

1. The authors mentioned "rubidium (Rb^+) possesses a smaller ionic radius compared with cesium (Cs^+)". The ionic radius values of Rb and Cs should be provided in the manuscript.
2. The low PLQY of 0.7% and ultra-fast PL lifetime of ~ 0.4 ns for $\text{Rb}_{0.6}\text{Cs}_{0.4}\text{PbBr}_3$ perovskite are attributed to a strong trap mediated non-radiative recombination. Is it still a pure phase for $\text{Rb}_{0.6}\text{Cs}_{0.4}\text{PbBr}_3$? Further discussion and/or evidence are needed for the low PLQY.
3. In Figure 1d, the authors described "uniform intensities along the Debye-Scherrer rings were observed in GIWAXS image, and indicating the quasi-2D perovskite crystallites are isotropically

oriented in the film". That's true. However, the treatment of the GIWAXS image is not fully correct. Usually, when images are displaced in the qx (parallel scattering vector) vs qz (vertical scattering vector), a missing wedge region should be marked.

4. The performance of the $\langle n \rangle = 3$ perovskite was significantly improved after introducing excess RbBr. The authors proposed the enhancement was derived from the "composition engineering" and "in-situ passivation". However, the mechanism seems not well described. For example, the authors described the XRD patterns of RbBr-rich $\langle n \rangle = 3$ perovskite "possesses very similar diffraction pattern as the pristine one" in Supplementary Figure 8. However, the XRD pattern shows a largely increased intensity at 28° compared with the pristine analogue. Please give a reasonable explanation.

5. The authors claimed that the performance of pristine $\langle n \rangle = 3$ could be largely improved by the treatment of excess RbBr, and the peak EQE was obtained with 100% excess RbBr. However, the description, characterization and analysis of the scenario of 120% and 140% are not enough. The authors should provide more detailed information for these cases.

6. The bleaching peak of $\langle n \rangle = 3$ species is not detected in the TA spectra, why?

8. The full name for the acronyms of PLQY, EQE, and T50 in the abstract should be given.

9. Supplementary Figs 11, 12, 17 and 18 are not mentioned in the manuscript. "Supplementary Figure 23" in Supplementary information should be "Supplementary Figure 19".

Reviewer #3 (Remarks to the Author):

This study presented that Rb-Cs alloyed quasi-2D perovskite structures could be a potential candidate material for stable blue perovskite light emitting diodes. The authors showed good optoelectronic properties of proposed Rb-Cs mixed quasi-2D perovskites with various analyzing methods including TA TRPL, and defect analysis. The performance of the proposed device seems to be good as both the stability and the efficiency are comparable to the state-of-the-art values. This report cannot attract broad interests if the best condition of the concentration and the number of layers cannot be changed. The authors suggested magic numbers of the concentration and the number of layers without proper validation. While this study contains several interesting points, I think this manuscript is rather premature to be published in nature communications due to the following reasons.

1) The authors should suggest the proper reason and detailed explanation why the $\langle n \rangle = 3$ perovskite structure can exhibit the best performance. The present manuscript contains only the results, and there is no physical origin and mechanism study the effects of the number of layers on the electronic structures and defect characteristics.

2) page6, line 102

The authors presented that XRD patterns of quasi-2D structures show different patterns with those of 3D analogous without verification. It is better to explain what is the origin of the difference. For example, density functional calculations can be helpful.

3) Optical band gaps of Rb-Cs mixed quasi-2D perovskites were modulated by the number of layer thickness and Rb contents. It is better to represent the band gap transition about the number of layers and Rb contents in one graph to enhance the readability.

Reviewer #1:

In this manuscript, the authors reported the synthesis of Rb-Cs alloyed perovskite quantum wells for pure-blue LED. The resulted perovskite film and LED device exhibited improved performance in terms of spectral stability, which is a well-known challenge for blue perovskite LED devices. They provide a new concept to construct blue perovskite emitter. This work will be interesting to perovskite community. Thus, the reviewer suggests publishing in *Nature Communications* after a major revision. More detailed comments are as follows.

Comment 1. The authors claimed the addition of excess amount of RbBr could significantly increase the PLQY, and the 100% excess of RbBr lead to the highest PLQY. It seems the PLQY start to decline the mount of RbBr more than 100%. Could the author provide the explanation?

Response: Thanks for the reviewer's comment. We have now provided the steady-state PL spectra, XRD patterns and TAS analysis, as well as PL lifetime for $\langle n \rangle_{\text{Rb0.6}} = 3$ perovskites with different amount of excess RbBr, to better understand the PLQY decline.

1). From the steady-state PL spectra (**Fig. S16d**), we confirmed the PL intensity increased with increasing amount of RbBr until 100% excess. Then, PL intensity started to decline with more RbBr introduced. The phenomenon was in good agreement with the evolutionary trend of PLQYs.

Fig. S16d Steady-state PL spectra of $\langle n \rangle_{\text{Rb0.6}} = 3$ perovskites with different content of excess RbBr.

2). As shown in the XRD pattern (**Fig. S12a**), the crystallinity of $\langle n \rangle_{\text{Rb}_{0.6} = 3}$ perovskites were significantly reduced when the amount of RbBr is beyond 100% excess, indicating that too much RbBr did inhibit the crystallization.

Fig. S12a XRD patterns of $\langle n \rangle_{\text{Rb}_{0.6} = 3}$ perovskites with different content of excess RbBr.

3). It is demonstrated that the lower crystallinity could induce a large number of defect states emerging (Wang, H. *et al. Nat. Commun.*, **2019**, 10, 665; Ball, J. M. *et al. Nat. Energy*, **2016**, 1, 16149). TAS and PL lifetime measurements were conducted to evaluate the trap densities of the resulting perovskites (**Fig. S17**). In particular, with the amount of excess RbBr exceeded 100%, the perovskite displayed almost one-order of magnitude higher trap density (**Fig. S17a**), and the PL lifetime was also reduced from ~ 21.2 to 18.7 and 15.1 ns, respectively (**Fig. S17b**), indicating a mass of defect states emerging during the process.

Fig. S17 (a) TAS and (b) PL lifetime for $\langle n \rangle_{\text{Rb}_{0.6} = 3}$ perovskites with different content of excess RbBr.

In brief, we now confirmed that too much RbBr (>100% excess) did inhibit the perovskites' crystallization, thus subsequently introduced a large number of defect states, leading to the PLQY decline.

Action: We have now provided the steady-state PL spectra, XRD patterns and TAS analysis as well as PL lifetime for $\langle n \rangle_{\text{Rb}0.6} = 3$ perovskites with different content of excess RbBr in **Supplementary Information Figs S16, S12 and S17**.

Comment 2. Why a 10 nm red-shift in PL spectra of extra RbBr sample compared to the pristine $\langle n \rangle_{\text{Rb}0.6} = 3$ perovskite sample? Did the extra RbBr changed the ratio of Rb:Cs in the perovskite crystal?

Response: Thanks for the reviewer's comment. We have now confirmed the Rb:Cs ratio do not be changed, and we now provided the XRD patterns and UV-vis absorption spectra of $\langle n \rangle_{\text{Rb}0.6} = 3$ perovskites with different content of excess RbBr for explanation.

1). As described in the manuscript (**Page 9**), the excess RbBr could induce the n -value shift to the larger numbers, and subsequently formed new perovskite species ($n = 5$ and $n > 5$). Then the radiative recombination will take place in the new emerged $n = 5$ and $n > 5$ domains, leading to the 10 nm red shift (Yuan, M. *et al. Nat. Nanotechnol.*, **2016**, 11, 872; Wang, N. *et al. Nat. Photonics*, **2016**, 10, 699). Similar phenomenon was also demonstrated in the previous work (Zou, W. *et al. Nat. Commun.*, **2018**, 9, 608)

2). As shown, the XRD peaks show no obvious shift among the samples with different amount of RbBr, demonstrating that the lattice structure remains unchanged (**Fig. S12**).

Fig. S12 (a) XRD patterns of $\langle n \rangle_{\text{Rb}0.6} = 3$ perovskites, and (b) detailed position comparison of the XRD peaks.

3). The exciton peaks in UV-vis absorption spectra (**Fig. S16c**) as well as GSB peaks in TA measurements (**Fig. 2**) for $\langle n \rangle_{\text{Rb}0.6} = 3$ perovskites exhibited no shifting

when different amount of excess RbBr introduced. Hence, the optical data confirmed the ratio of Rb to Cs was unchanged during the addition of excess RbBr.

Fig. S16c UV-vis absorption of $\langle n \rangle_{\text{Rb0.6}} = 3$ with different content of excess RbBr.

Fig. 2 TA measurements for (a, b) pristine and (c, d) RbBr-rich $\langle n \rangle_{\text{Rb0.6}} = 3$ perovskites.

In brief, we confirmed excess amount of RbBr could regulate the n -value distribution and induce large n -value domains emerging. Thus, red-shift of the PL spectra was observed due to the energy transfer. From XRD patterns, UV-vis absorption and TA spectra, we now confirmed the ratio of Rb to Cs remain unchanged.

Action: We have now provided XRD patterns and UV-vis absorption spectra of $\langle n \rangle_{\text{Rb}0.6} = 3$ perovskites with different content of excess RbBr in **Supplementary Information Figs S12, S16c.**

Comment 3. The authors studied the stability of perovskite film after 10 day (SI Figure 10). How about the PLQY of the perovskite film after 10 days.

Response: Thanks for the reviewer's comment. We have now provided the PLQY tracking data in **Fig. S30**. The data shows PLQY of RbBr-rich $\langle n \rangle_{\text{Rb}0.6} = 3$ was only slightly decreased from around 82% to 79% after 10 days ageing.

Fig. S30 PLQY tracking data for quasi-2D perovskite films.

Action: We have now provided the PLQY tracking data of the perovskite films in **Supplementary Information Fig. S30.**

Comment 4. In the transient absorption (TA) measurements, the authors claimed that the $n \gg 5$ perovskite domains could be produced in pristine $\langle n \rangle = 5$ perovskite, and possess higher defect density which is similar with 3D perovskite. Thus, the measurement on 3D $\text{Rb}_{0.6}\text{Cs}_{0.4}\text{PbBr}_3$ perovskite should be also provided to confirm the hypothesis.

Response: Thanks for the comment. We have now provided TA measurement on 3D $\text{Rb}_{0.6}\text{Cs}_{0.4}\text{PbBr}_3$ perovskite (**Fig. S4**). We observed an Urbach energy tail in the TA spectra, which indicate serious energy disorder should exist in the 3D perovskite.

Consequently, we now confirmed the hypothesis and proved the 3D perovskite possessed high defect density.

As shown in **Fig. S4**, the bleaching edge of 3D $\text{Rb}_{0.6}\text{Cs}_{0.4}\text{PbBr}_3$ perovskite is around 495 nm, which is consistent with the result of UV-vis absorption spectra. We also carried out the band-tail measurement from the transient bleaching, and found a ~ 9.2 meV Urbach tail in it, demonstrating a largely energy disorder derived from serious defect state in the film, which could result in significant non-radiative recombination (Wang, H. *et al. Nat. Commun.*, **2019**, 10, 665; Shao, Y. *et al. Nat. Energy*, **2016**, 1, 15001).

Fig. S4 TA measurement for 3D perovskite $\text{Rb}_{0.6}\text{Cs}_{0.4}\text{PbBr}_3$.

Action: We have now provided TA measurement on 3D $\text{Rb}_{0.6}\text{Cs}_{0.4}\text{PbBr}_3$ perovskite in **Supplementary Information Fig. S4**.

Comment 5. In steady state absorption, there are four excitonic absorption peaks $n = 1, 2, 3, 4$, why did the TA feature of $n = 3$ phase absent?

Response: Thanks for the reviewer's valuable comment. We now confirmed the closed packed GSB peaks and the ESA signal observed around 440 nm may account for the reasons that leading to unapparent $n = 3$ feature.

1). As shown, the GSB peaks of $n = 1, 2, 3$ and 4 species could be easily distinguished in pristine $\langle n \rangle_{\text{Rb}0.4} = 3$ perovskite (**Fig. S20a, b**). However, with the increasing ratio of Rb to Cs, the bandgap difference between each n species become smaller, leading to the resulting GSB peaks closely packed. Thus, the larger n -value GSB peaks could hide from each other and become difficult to be distinguished (Li,

M. *et al. Adv. Funct. Mater.*, **2018**, 28, 1707006). For instance, in pristine $\langle n \rangle_{\text{Rb}0.8} = 3$ perovskite, the $n = 3$ and 4 species could form a broad and continuous bleaching (Fig. S20e, f).

Fig. S20 TA measurements for (a, b) pristine $\langle n \rangle_{\text{Rb}0.4} = 3$ and (e, f) pristine $\langle n \rangle_{\text{Rb}0.8} = 3$ perovskites.

2). We now provided the TA spectrum for pristine $\langle n \rangle_{\text{Rb}0.6} = 2$ perovskite, and confirmed the GSB peak for $n = 3$ species is located around 445 nm (Fig. S19c, d), which is consistent with the results from UV-vis absorption spectra (Fig. S16c).

3). In addition, due to the complicated electronic coupling process among different n species, a strong excited state absorption (ESA) signal was found at $\Delta\text{OD} > 0$ region. The $n = 3$ GSB peaks of pristine $\langle n \rangle_{\text{Rb}0.6} = 3$ and RbBr-rich $\langle n \rangle_{\text{Rb}0.6} = 3$ were also hidden by these ESA signals, and became more difficult to be pointed out (Shang, Q. *et al. J. Phys. Chem. Lett.*, **2017**, 8, 4431).

4). However, we still observed some sags at around 445 nm in ESA statuses for pristine and RbBr-rich $\langle n \rangle_{\text{Rb}0.6} = 3$, which should from $n = 3$ GSB peaks (Fig. S19e, f).

Fig. S19 (c, d) TA measurements for pristine $\langle n \rangle_{\text{Rb0.6}} = 2$ perovskite. (e, f) Enlarged TA spectra at selected timescales for pristine and RbBr-rich $\langle n \rangle_{\text{Rb0.6}} = 3$ perovskites.

In brief, we found the GSB peaks for different n -value species closely packed together, and then become difficult to be distinguished. In addition, strong ESA signals at ~ 440 nm was observed, which might significantly influence the signal of $n = 3$ GSB peak.

Action: We have now provided TA measurements on pristine $\langle n \rangle_{\text{Rb0.4}} = 3$, $\langle n \rangle_{\text{Rb0.8}} = 3$ and $\langle n \rangle_{\text{Rb0.6}} = 2$ perovskites in **Supplementary Information Figs S19, S20**. The relevant discussion was provided in **Supplementary Information Note S3**.

Comment 6. In order to make the device performance data more convincing, the authors need to provide statistical data.

Response: Thanks so much for the reviewer's suggestion. We have now provided the histograms of maximum luminance and EQEs for pristine and RbBr-rich $\langle n \rangle_{\text{Rb0.6}} = 3$ based devices (**Fig. S26**).

Fig. S26 Histograms of (a, c) max luminance and (b, d) EQEs for pristine and RbBr-rich $\langle n \rangle_{\text{Rb0.6}} = 3$ based devices.

Action: We have now provided histograms of maximum luminance, EQEs for pristine and RbBr-rich $\langle n \rangle_{\text{Rb0.6}} = 3$ based devices in **Supplementary Information Fig. S26**.

Comment 7. Although the lifetime of perovskite LED was improved, 14.5 min is still in the same region as other reports.

Response: Thanks for the reviewer's comment.

Last year, the very stable sky-blue PeLED was reported by controlling the n values distribution of quasi-2D perovskites (Xing, J. *et al. Nat. Commun.*, **2018**, 9, 3541), representing a significant breakthrough in this field. We have now acknowledged this breakthrough in the revised manuscript (**Page 4**).

In our work, we provided a new strategy to obtain spectra-stable blue emitting perovskite films with relatively high PLQYs. The strategy focuses on engineering the "A-site" cation. Encouraged by these findings, we realized the emitters in the whole blue spectral region (from 454 to 492 nm) by further regulating the composition of Rb-Cs alloyed quasi-2D perovskites (**Fig. S28**). Thus, we believe that the strategy in our manuscript could also provide a choice to meet the various purpose in PeLED fields.

Fig. S28 PL emission wavelengths and corresponding PLQYs of $\langle n \rangle_{Rb0.8} = 3$, $\langle n \rangle_{Rb0.6} = 3$ and $\langle n \rangle_{Rb0.4} = 3$ perovskites with different content of excess RbBr.

Action: We have now acknowledged the breakthrough that made by Xing and co-authors in the manuscript. We also provided more % data to demonstrate versatile application of our strategy, and the discussion is provided in **Supplementary Information Note S4**.

Comment 8. The author should obtain some help in polishing the language, there are many grammatical errors, or inappropriate statement. Particularly, there are many exaggerated adjectives through the manuscript, such as superfast, terrible, etc., and missing "that" or "which" for clauses.

Response: Thanks for the comment. We have revised the manuscript thoroughly, and tried to avoid any grammar or syntax error as well as colloquial terms to meet the high criteria of *Nature communications*.

Action: The manuscript has been revised thoroughly.

Reviewer #2:

In the manuscript, Jiang *et al.* report the fabrication of spectra stable and efficient blue perovskite light-emitting diodes. After introducing the Rb into the precursor system, the reduced-dimensional Rb-Cs alloyed perovskite is formed. Quasi-2D structure was proved to be able to stabilize the metastable perovskite phase and facilitate the energy transfer. Detailed energy transfer/combination kinetics was carefully investigated through transient spectroscopy. After in-situ passivation, a high EQE of 1.35% with excellent spectra stability was obtained. Overall, this work provides a new strategy to generate spectra stable blue perovskite LED. The paper is interesting and will be helpful to researchers in related fields. Therefore, I recommend the paper published in *Nature Communications* after minor revision as follows:

Comment 1. The authors mentioned “rubidium (Rb^+) possesses a smaller ionic radius compared with cesium (Cs^+)”. The ionic radius values of Rb and Cs should be provided in the manuscript.

Response: Thanks for the reviewer’s suggestion. The cationic radius values of Rb (152 pm) and Cs (167 pm) have been provided in our revised manuscript according to previous research (Saliba, M. *et al. Science*, **2016**, 354, 6309).

Action: The cationic radius values of Rb and Cs have been provided in our revised manuscript (**Paragraph 2, Page 5**).

Comment 2. The low PLQY of 0.7% and ultra-fast PL lifetime of ~ 0.4 ns for $\text{Rb}_{0.6}\text{Cs}_{0.4}\text{PbBr}_3$ perovskite are attributed to a strong trap mediated non-radiative recombination. Is it still a pure phase for $\text{Rb}_{0.6}\text{Cs}_{0.4}\text{PbBr}_3$? Further discussion and/or evidence are needed for the low PLQY.

Response: Thanks for the comment. We now confirmed $\text{Rb}_{0.6}\text{Cs}_{0.4}\text{PbBr}_3$ is a pure phase from XRD measurement. We have now provided the TA analysis for 3D perovskite $\text{Rb}_{0.6}\text{Cs}_{0.4}\text{PbBr}_3$ to better understand the reasons leading to low PLQY.

1). The XRD pattern of fresh $\text{Rb}_{0.6}\text{Cs}_{0.4}\text{PbBr}_3$ is shown in **Fig. S2b**. The main diffraction peaks at around 16° and 32° can be attributed to the (100) and (200) planes, corresponding to the orthorhombic perovskite phase (Linaburg, M. R., *et al. Chem. Mater.*, **2017**, 29, 3507). No obvious peak is indexable to any impurity phase. Thus, we confirmed $\text{Rb}_{0.6}\text{Cs}_{0.4}\text{PbBr}_3$ is a pure phase.

Fig. S2b XRD patterns of 3D perovskites $\text{Rb}_x\text{Cs}_{1-x}\text{PbBr}_3$ with different values of “x”

2). We observed an obvious band-tail (Urbach tail) around 9.2 meV in TA spectrum of $\text{Rb}_{0.6}\text{Cs}_{0.4}\text{PbBr}_3$ perovskite, indicating large energy disorder and huge defect states was existed in $\text{Rb}_{0.6}\text{Cs}_{0.4}\text{PbBr}_3$ (Wang, H. *et al. Nat. Commun.*, **2019**, 10, 665; Shao, Y. *et al. Nat. Energy*, **2016**, 1, 15001).

Fig. S4 TA measurement for 3D perovskite $\text{Rb}_{0.6}\text{Cs}_{0.4}\text{PbBr}_3$.

In summary, we confirmed the $\text{Rb}_{0.6}\text{Cs}_{0.4}\text{PbBr}_3$ is a pure phase, however, a mass of defect state still exists in this perovskite, resulting in a low PLQY and fast PL lifetime.

Action: We have now provided the TA measurement on 3D perovskite $\text{Rb}_{0.6}\text{Cs}_{0.4}\text{PbBr}_3$ in **Supplementary Information Fig. S4**, and relative discussion was added **Supplementary Information Page 5**.

Comment 3. In Figure 1d, the authors described “uniform intensities along the Debye-Scherrer rings were observed in GIWAXS image, and indicating the quasi-2D perovskite crystallites are isotropically oriented in the film”. That’s true. However, the treatment of the GIWAXS image is not fully correct. Usually, when images are displaced in the q_{xy} (parallel scattering vector) vs q_z (vertical scattering vector), a missing wedge region should be marked.

Response: Thanks for the reviewer’s suggestion. We have now updated the GIWAXS image (**Fig. 1d**) in the revised version.

Fig. 1d GIWAXS pattern of RbBr-rich $\langle n \rangle_{\text{Rb}0.6} = 3$ film.

Action: The GIWAXS image was updated in the revised manuscript (**Fig. 1d**).

Comment 4. The performance of the $\langle n \rangle = 3$ perovskite was significantly improved after introducing excess RbBr. The authors proposed the enhancement was derived from the “composition engineering” and “in-situ passivation”. However, the

mechanism seems not well described. For example, the authors described the XRD patterns of RbBr-rich $\langle n \rangle_{\text{Rb}0.6} = 3$ perovskite “possesses very similar diffraction pattern as the pristine one” in Supplementary Figure 8. However, the XRD pattern shows a largely increased intensity at 28° compared with the pristine analogue. Please give a reasonable explanation.

Response: Thanks for the comment. We have now provided the XRD patterns for $\langle n \rangle_{\text{Rb}0.6} = 3$ with different amount of excess RbBr (from 0% to 140%) in order to better understand the passivation effect.

As shown, three dominant diffraction peaks could be observed in $\langle n \rangle_{\text{Rb}0.6} = 3$ samples, and the diffraction peak at around 28° can be indexed to crystallographic plane (202) of quasi-2D perovskite (Tsai, H. *et al. Nature*, **2016**, 536, 312). With the treatment of excess RbBr, the intensity of (202) continuously enhanced, indicating the crystallinity along (202) plane of RbBr-rich $\langle n \rangle_{\text{Rb}0.6} = 3$ is much higher than pristine analogue. The highly oriented structure reduced the disorder of the crystal and then suppressed the defect state formation (Wang, H. *et al. Nat. Commun.*, **2019**, 10, 665). Thus, the passivation effect of excess RbBr is pluralism, and the excess RbBr can also passivate crystal defect by affecting the crystallization process as well as crystal orientation.

Fig. S12a XRD patterns of $\langle n \rangle_{\text{Rb}0.6} = 3$ perovskites with different content of excess RbBr.

Action: We have now provided the XRD patterns for $\langle n \rangle_{\text{Rb}0.6} = 3$ with different amount of excess RbBr (from 0% to 140%) in **Supplementary Information Fig. S12a**.

Comment 5. The authors claimed that the performance of pristine $\langle n \rangle = 3$ could be largely improved by the treatment of excess RbBr, and the peak EQE was obtained with 100% excess RbBr. However, the description, characterization and analysis of the scenario of 120% and 140% are not enough. The authors should provide more detailed information for these cases.

Response: Thanks so much for your suggestion. We have now added steady-state PL spectra, XRD patterns, defect state analysis, and top-view SEM images for $\langle n \rangle_{\text{Rb0.6}} = 3$ perovskites with different content of excess RbBr to better understand the role of excess RbBr on the properties of materials.

1). We now confirmed the PL intensity enhanced with the increasing content of excess RbBr until 100% (**Fig. S16d**), and started to decline while the content of excess RbBr was higher, which is consistent with the results of PLQYs.

Fig. S16d Steady-state PL spectra of $\langle n \rangle_{\text{Rb0.6}} = 3$ perovskites with different content of excess RbBr.

2). We now observed the crystallinity of Rb-Cs alloyed quasi-2D perovskites was significantly reduced while the content of excess RbBr exceeded 100%, demonstrating that too much RbBr did inhibit the crystallization (**Fig. S12a**).

Fig. S12a XRD patterns of $\langle n \rangle_{\text{Rb0.6}} = 3$ perovskites with different content of excess RbBr.

3). It is well established that the lower crystallization could induce a large number of defect states. (Wang, H. *et al. Nat. Commun.*, **2019**, 10, 665; Ball, J. M. *et al. Nat. Energy*, **2016**, 1, 16149). Thus, TAS and PL lifetime measurements were carried out to evaluate the trap densities in the resulting perovskites. In brief, while the amount of excess RbBr is above 100%, the PL lifetime was reduced from ~ 21.2 to 18.7 and 15.1 ns, respectively, indicating a serious non-radiative recombination (**Fig. S17b**). Meanwhile the perovskite also displayed almost one-order of magnitude higher trap density (**Fig. S17a**), which is consistent with the results of PL lifetime. Thus, we now confirmed that the too much RbBr did inhibit the crystallization and then formed a large number of defect states, resulting in reduced PLQYs.

Fig. S17 (a) TAS, and (b) PL lifetime for $\langle n \rangle_{\text{Rb0.6}} = 3$ perovskites with different content of excess RbBr.

4). Furthermore, top-view SEM images of $\langle n \rangle_{\text{Rb0.6}} = 3$ perovskites with different content of excess RbBr were also carried out (**Fig. S18**). As shown, too much RbBr could induce significant deterioration of the morphology.

Fig. S18 Top-view SEM images of the $\langle n \rangle_{\text{Rb0.6}} = 3$ perovskites with different content of excess RbBr.

Action: We have now provided the steady-state PL spectra, XRD patterns, TAS analysis, PL lifetime and top-view SEM for $\langle n \rangle_{\text{Rb}0.6} = 3$ perovskites in **Supplementary Information Figs S16, S12, S17 and S18.**

Comment 6. The bleaching peak of $\langle n \rangle = 3$ species is not detected in the TA spectra, why?

Response: Thanks so much for the reviewer's comment. We now confirmed the closed packed GSB peaks as well as the ESA signal observed around 440 nm make out the reasons for the unobvious of $n = 3$ feature.

1). As shown, the GSB peaks of $n = 1, 2, 3$ and 4 could be easily distinguished in pristine $\langle n \rangle_{\text{Rb}0.4} = 3$ perovskite (**Fig. S20a, b**). As the ratio of Rb to Cs increased, the bandgap difference between each n species become smaller, resulting in a closed packed GSB peaks. Thus, the larger n -value GSBs peaks could hide from each other and become difficult to be distinguished. For instance, in pristine $\langle n \rangle_{\text{Rb}0.8} = 3$ perovskite, the GSB peaks of $n = 3$ and 4 species could form an extensive and continuous bleaching (Fig. S20e, f).

Fig. S20 TA measurements for (a, b) pristine $\langle n \rangle_{\text{Rb}0.4} = 3$ and (e, f) $\langle n \rangle_{\text{Rb}0.8} = 3$ perovskites.

2). Furthermore, we also investigated the TA measurement on pristine $\langle n \rangle_{\text{Rb0.6}} = 2$ and confirmed the GSB peak of $n = 3$ species is located around 445 nm (**Fig. S19c, d**), which is in agree with the results from UV-vis absorption spectra (**Fig. S16c**).

3). In addition, a strong excited state absorption (ESA) signal was found at $\Delta \text{OD} > 0$ region owing to the complex coupling process among different n species (Shang, Q. *et al. J. Phys. Chem. Lett.*, **2017**, 8, 4431). The $n = 3$ GSB peaks of pristine $\langle n \rangle_{\text{Rb0.6}} = 3$ and RbBr-rich $\langle n \rangle_{\text{Rb0.6}} = 3$ were also hidden by these ESA signals, and became more difficult to be pointed out.

4). However, there are also distinct sags could be found in ESA status at around 445 nm for pristine $\langle n \rangle_{\text{Rb0.6}} = 3$ and RbBr-rich $\langle n \rangle_{\text{Rb0.6}} = 3$, probably deriving from $n = 3$ GSB peaks (**Fig. S19e, f**).

Fig. S19 (c, d) TA measurements for pristine $\langle n \rangle_{\text{Rb0.6}} = 2$ perovskite. (e, f) Enlarged TA spectra at selected timescales of pristine and RbBr-rich $\langle n \rangle_{\text{Rb0.6}} = 3$ perovskites.

Fig. S16c UV-vis absorption of $\langle n \rangle_{\text{Rb0.6}} = 3$ with different content of excess RbBr.

In brief, GSB peaks for different n species closely packed together and become difficult to distinguish. Moreover, strong ESA signals were observed at ~ 440 nm, and could significantly influence the signal of $n = 3$ species GSB peak.

Action: We have carried out TA measurements on pristine $\langle n \rangle_{\text{Rb0.4}} = 3$, $\langle n \rangle_{\text{Rb0.8}} = 3$, and $\langle n \rangle_{\text{Rb0.6}} = 2$ perovskites in **Supplementary Information Figs S19, S20**. The relevant discussion was provided in **Supplementary Information Note S3**.

Comment 7. The full name for the acronyms of PLQY, EQE, and T_{50} in the abstract should be given.

Response: Thanks for the suggestion. We have added the full name of PLQY (high photoluminescence quantum yield), EQE (external quantum efficiency) and T_{50} (half-lifetime) in the abstract.

Action: We have added the full name of all acronyms in the revised version.

Comment 8. Supplementary Figs 11, 12, 17 and 18 are not mentioned in the manuscript. “Supplementary Figure 23” in Supplementary information should be “Supplementary Figure 19”.

Response: Thanks for the suggestion. We have now added the relative descriptions of all Figures in our revised manuscript, and ensure the correction of the numbers for all Figure Captions.

Action: The Supplementary Information Figs S11, S12, S17 and S18, have been mentioned in our revised manuscript, and the “Supplementary Figure 23” has been modified in correct order.

Reviewer #3:

This study presented that Rb-Cs alloyed quasi-2D perovskite structures could be a potential candidate material for stable blue perovskite light emitting diodes. The authors showed good optoelectronic properties of proposed Rb-Cs mixed quasi-2D perovskites with various analyzing methods including TA, TRPL, and defect analysis. The performance of the proposed device seems to be good as both the stability and the efficiency are comparable to the state-of-the-art values. This report cannot attract broad interests if the best condition of the concentration and the number of layers cannot be changed. The authors suggested magic numbers of the concentration and the number of layers without proper validation. While this study contains several interesting points, I think this manuscript is rather premature to be published in nature communications due to the following reasons.

Response: We really appreciated the reviewer's insightful comments. **We now provided more data to show the versatile applications of this strategy. We now were able to realize blue perovskite emitters in the entire blue-spectral region, and we now provided a sky-blue PLEDs with EQE of 1.48 % and T_{50} lifetime of 18.7 min using RbBr-rich $\langle n \rangle_{\text{Rb}0.4} = 3$ without systematic optimization.**

1). Efficient and spectral stable pure-blue LEDs with $\lambda_{\text{EL}} < 480$ nm is highly desired for display application, which is also highly challenging for the perovskite LED community. (Wu, Y. *et al. ACS Energy Lett.*, **2019**, 4, 673). Thus, we are aiming to realize efficient and stable pure-blue PLEDs in this manuscript initially.

As we know, the ratio of Rb: Cs, $\langle n \rangle$ -values and amount of RbBr influenced optical properties of the resulting perovskites coherently. We have now provided the PL wavelengths and PLQYs data for the alloyed quasi-2D perovskites with different compositions. As shown in **Fig. S28** and **Fig. S30**, after screening, we found the quasi-2D perovskite with the ratio of Rb: Cs = 0.6: 0.4 should be the most suitable candidate to realize stable pure-blue (< 480 nm) PeLEDs in consideration of emission wavelength, PLQYs values and materials' stabilities.

Fig. S28 PL emission wavelengths and corresponding PLQYs of $\langle n \rangle_{Rb0.8} = 3$, $\langle n \rangle_{Rb0.6} = 3$ and $\langle n \rangle_{Rb0.4} = 3$ perovskites with different content of excess RbBr.

Fig. S30 PLQY tracking data for quasi-2D perovskite films.

2). High color purity is another prerequisite for high quality PeLEDs.

After careful investigation, we found the $\langle n \rangle$ -value distribution would significantly influence the color purity (FWHM) as well as overall PLQYs. As shown in Fig. S7e, we are not interested in the low n -values ($n = 1, 2$) perovskites due to the extremely low PLQYs that originate from inefficient internal energy transfer (Yang, X. *et al. Nat. Commun.*, **2018**, 9, 570; Yuan, M. *et al. Nat. Nanotechnol.*, **2016**, 11, 872). Similarly, $\langle n \rangle = 5$ exhibited relatively lower PLQY compared to $\langle n \rangle = 3$ due to the higher defect density, which we have now briefly discussed in the manuscript (Page 6) and Supplementary Information (Page 41). However, both $\langle n \rangle = 4$ and $\langle n \rangle = 5$ perovskites exhibit broad FWHM due to the inefficient energy transfer caused by the flatten dispersed band structure. Broad FWHM obviously not fit for the high-quality LED application. Thus, after careful screening, in consideration of the

color purity, we select $\langle n \rangle = 3$ perovskites for further optimization to fabricate LEDs.

Fig. S7 (a) Absorption and (b) steady-state PL spectra; (c) low- and (d) wide-angle X-ray diffraction (XRD) patterns; and (e) PLQYs of $\text{PEA}_2(\text{Rb}_{0.6}\text{Cs}_{0.4})_{n-1}\text{Pb}_n\text{Br}_{3n+1}$ perovskites with different $\langle n \rangle$ -values. (f) TAS and (g) PL lifetime for pristine $\langle n \rangle_{\text{Rb}0.6} = 3, 4$ and 5 perovskites.

Following this strategy, after carefully optimization, we then were able to achieved the spectra-stable pure-blue PeLEDs. *Reviewer #1* also valued our Rb-Cs alloyed quasi-2D perovskite shows big advantages. He commented “*They provide a new concept to construct blue perovskite emitter. This work will be interesting to perovskite community.*”

Furthermore, thanks to the suggestion from the *Reviewer #3*, we now provided more data to show the versatile applications of this strategy. We now were able to realize blue perovskite emitters in the entire blue-spectral region, and we now provided a sky-blue PLEDs with EQE of 1.48 % and T_{50} lifetime of 18.7 min using $\langle n \rangle_{\text{Rb}0.4} = 3$ without systematic optimization:

3). As shown in **Figs S20, S16**, we now achieved perovskite blue emitters that covers the entire blue spectral region, from 454 to 492 nm, by tuning the composition (**Fig. S28**). In particular, RbBr-rich $\langle n \rangle_{\text{Rb}0.8} = 3$ (458 nm, deep-blue) and the RbBr-rich $\langle n \rangle_{\text{Rb}0.4} = 3$ (490 nm, sky-blue) exhibited reasonable PLQYs of 58.3 % and 84.1 %, respectively.

Fig. S20 TA measurements for pristine $\langle n \rangle_{\text{Rb}0.4} = 3$, $\langle n \rangle_{\text{Rb}0.8} = 3$ and RbBr-rich $\langle n \rangle_{\text{Rb}0.4} = 3$, $\langle n \rangle_{\text{Rb}0.8} = 3$ perovskites.

Fig. S16 (a, c, e) UV-vis absorption and (b, d, f) steady-state PL spectra of the perovskites with different compositions.

We then fabricated a sky-blue PeLED using the RbBr-rich $\langle n \rangle_{\text{Rb}0.4} = 3$ following the same strategy. The sky-blue PeLEDs exhibit a peak EQE of 1.48 %, peak luminance of 854.3 cd m^{-2} and T_{50} lifetime of 18.7 min without systematic

optimization (**Figs S31, S32**), representing one of the most stable sky-blue PeLEDs (Xing, J. *et al. Nat. Commun.*, **2018**, 9, 3541).

Fig. S31 (a) *I-V*, *V-L* curves; (b) *EQE-I* characteristic; (c) *EL* spectra; (d) *CIE* coordinate of $\text{RbBr-rich } \langle n \rangle_{\text{Rb}0.4} = 3$ based devices. (e, g) *EL* spectra change and (f, h) T_{50} measurements under max *EQE* point and constant voltage of 4.0 V of $\text{RbBr-rich } \langle n \rangle_{\text{Rb}0.4} = 3$ based devices.

Fig. S32 Histograms of (a) max luminance and (b) *EQEs* for $\text{RbBr-rich } \langle n \rangle_{\text{Rb}0.4} = 3$ based devices.

In summary, we herein reported a brand-new concept to achieve efficient and stable blue perovskite LEDs. We showed the “A-site” cation engineering strategy to be a versatile method to stabilize the perovskite and regulate the emission properties. This strategy now was confirmed to be able to generate efficient perovskite blue emitters and LEDs in the entire blue spectral region.

Action: We have now provided the defect state characteristics of pristine $\langle n \rangle_{\text{Rb}0.6} = 4$ and 5 in **Supplementary Information Fig. S7f, g**. The UV-vis absorption, steady-state PL and TA spectra of $\langle n \rangle_{\text{Rb}0.8} = 3$ and $\langle n \rangle_{\text{Rb}0.4} = 3$ perovskites were shown in **Supplementary Information Figs S16, S20 and Note S4**. The PL wavelengths and corresponding PLQYs as well as the PLQY tracking date of the

perovskites with different compositions were shown in **Supplementary Information Figs S28, S30**. We have also fabricated the sky-blue PeLEDs as shown in **Supplementary Information Figs S31, S32 and Note S5**.

Comment 1. The authors should suggest the proper reason and detailed explanation why the $\langle n \rangle = 3$ perovskite structure can exhibit the best performance. The present manuscript contains only the results, and there is no physical origin and mechanism study the effects of the number of layers on the electronic structures and defect characteristics.

Response: Thanks so much for your very valuable suggestion. We have now provided new characterizations for $\text{PEA}_2(\text{Rb}_{0.6}\text{Cs}_{0.4})_{n-1}\text{Pb}_n\text{Br}_{3n+1}$ with different $\langle n \rangle$ -values in **Fig. S7**. In particular, we have now provided the TAS analysis and PL lifetime measurements for $\langle n \rangle = 3, 4$ and 5 (**Fig. S7f, g**).

As shown in **Fig. S7e**, the $\langle n \rangle = 1$ and $\langle n \rangle = 2$ perovskites show extremely low PLQYs because of the inefficient internal energy transfer (Yang, X. *et al. Nat. Commun.*, **2018**, 9, 570; Yuan, M. *et al. Nat. Nanotechnol.*, **2016**, 11, 872). Thus, $\langle n \rangle = 3, 4$ and 5 seem to be the candidates. However, the $\langle n \rangle = 5$ with a PL emission at around 480 nm also exhibits a relatively low PLQY due to the higher defect density (**Fig. S7f, g**). Furthermore, despite of the $\langle n \rangle = 4$ exhibits a slightly higher PLQY, the PL spectra is much broad than that of $\langle n \rangle = 3$. Thus, considering of high efficiency and color purity, the $\langle n \rangle = 3$ could be a better candidate for the PeLEDs in our system.

Furthermore, we also carried out the photocarrier transportation and recombination dynamics on pristine $\langle n \rangle_{\text{Rb0.6}} = 3$ and 5 in our manuscript. We found that the unreasonable n species distribution could significantly block the carrier transfer process, resulting in PL spectral broadening. Moreover, especially for $\langle n \rangle = 5$, the larger- n values ($n \gg 5$) perovskite domain was found and could form non-radiative recombination center, leading to a PLQY decline, consistent with the

results of TAS and PL lifetime. Thus, the modulation of the n distribution should be more precisely.

Fig. S7 (a) Absorption and (b) steady-state PL spectra; (c) low- and (d) wide-angle X-ray diffraction (XRD) patterns; and (e) PLQYs of $\text{PEA}_2(\text{Rb}_{0.6}\text{Cs}_{0.4})_{n-1}\text{Pb}_n\text{Br}_{3n+1}$ perovskites with different $\langle n \rangle$ -values. (f) TAS and (g) PL lifetime for pristine $\langle n \rangle_{\text{Rb}0.6} = 3, 4$ and 5 perovskites.

Action: We have now provided the characteristics for $\text{PEA}_2(\text{Rb}_{0.6}\text{Cs}_{0.4})_{n-1}\text{Pb}_n\text{Br}_{3n+1}$ with different $\langle n \rangle$ -values in **Supplementary Information Fig. S7** and detailed discussion was added in **Supplementary Information Note S1**. Specially, we also investigated the defect state for $\langle n \rangle_{\text{Rb}0.6} = 3, 4$ and 5 by TAS and PL lifetime in **Supplementary Information Fig. S7f, g**.

Comment 2. Page 6, line 102. The authors presented that XRD patterns of quasi-2D structures show different patterns with those of 3D analogous without verification. It is better to explain what is the origin of the difference. For example, density functional calculations can be helpful.

Response: Thanks for the reviewer's suggestion. We now observed the (0 0 k) diffraction peaks from the XRD patterns, which represent the fingerprint peak for

quasi-2D structure. We have now provided the XRD patterns in low-diffraction-angle, and calculated the lattice/unit constants from XRD patterns by Bragg diffraction equation. We also added the simplified schematic representation for the structure of $\text{PEA}_2(\text{Rb}_{0.6}\text{Cs}_{0.4})_{n-1}\text{Pb}_n\text{Pb}_{3n+1}$.

As shown, the (0 0 k) diffraction peaks derived from $n = 1$ and 2 species could be observed in $\langle n \rangle = 1$ and $\langle n \rangle_{\text{Rb}_{0.6}} = 2$ perovskites respectively (**Fig. S7**), demonstrating the formation of layered structure. Meanwhile, in the low-diffraction-angle region (**Fig. S7c**), the (0 0 k) diffraction peak could also be found in $\langle n \rangle_{\text{Rb}_{0.6}} = 2$ and $\langle n \rangle_{\text{Rb}_{0.6}} = 3$ perovskites respectively, confirmed the quasi-2D structure. With the n -values increasing, the diffraction peak of low n species disappeared, probably due to very weak small n phases diffractions (Yang, X. *et al. Nat. Commun.*, **2018**, 9, 570; Xiao, Z. *et al. Nat. Photonics*, **2017**, 11, 108).

Fig. S7 (c) low- and (d) wide-angle X-ray diffraction (XRD) patterns.

In order to further compare the crystal structure of 3D and quasi-2D perovskites, we also added the simplified schematic representation for the crystal structure in **Fig. S8**, and the relevant lattice/unit constants were also calculated from the XRD patterns by Bragg diffraction equation (**Table S2**). As shown, the (0 0 k) diffraction peaks were observed in $\langle n \rangle = 1$ perovskite with a layer lattice constant (d) of 16.8 Å, indicating the formation of 2D ($n = 1$ species) perovskite (Kitazawa, N. *et al. J. Phys. Chem. Solids*, **2010**, 71, 797). Furthermore, it can also be calculated that the d of 3D perovskite $\text{Rb}_{0.6}\text{Cs}_{0.4}\text{PbBr}_3$ is 5.6 Å from the diffraction peak of (1 0 0) and (2 0 0). As

expected, the d of $\langle n \rangle = 2$ is measured to be 22.4 Å, which fully fits the expected $n = 2$ quasi-2D layered perovskite structure ($16.8 \text{ Å} + (n - 1) \times 5.6 \text{ Å}$, $n = 2$).

Fig. S8 Schematic representation of the crystal structures of $\text{PEA}_2(\text{Rb}_{0.6}\text{Cs}_{0.4})_{n-1}\text{Pb}_n\text{Pb}_{3n+1}$ ($n = 1, 2$ and ∞).

Table S2 Parameters of the XRD patterns.

Composition	XRD peak	2θ (°)	θ (°)	m	d (Å)
$\text{Rb}_{0.6}\text{Cs}_{0.4}\text{PbBr}_3$	(1 0 0)	15.58	7.79	1	5.6
	(2 0 0)	31.77	15.89	2	5.7
Pristine $\langle n \rangle = 1$	(0 0 2)	5.29	2.65	1	16.7
	(0 0 4)	10.47	5.23	2	16.8
	(0 0 6)	15.71	7.86	3	16.8
	(0 0 8)	21.06	10.53	4	16.8
	(0 0 10)	26.46	13.23	5	16.8
	(0 0 12)	31.94	15.97	6	16.8
Pristine $\langle n \rangle_{\text{Rb0.6}} = 2$	(0 0 4)	7.88	3.94	2	22.4
	(0 0 6)	11.84	5.92	3	22.5
	(0 0 8)	15.77	7.89	4	22.4
	(0 0 10)	19.69	9.85	5	22.5
	(0 0 14)	27.4	13.70	7	22.5
	(0 0 16)	31.77	15.89	8	22.4

Action: We have provided the XRD patterns in low-diffraction-angle (Supplementary Fig. S7c), and calculated the lattice/unit constants from XRD patterns (Supplementary Information Table S2). We also added the schematic representation for the crystal structure of $\text{PEA}_2(\text{Rb}_{0.6}\text{Cs}_{0.4})_{n-1}\text{Pb}_n\text{Pb}_{3n+1}$ in Supplementary Information Fig. S8.

Comment 3. Optical band gaps of Rb-Cs mixed quasi-2D perovskites were modulated by the number of layer thickness and Rb contents. It is better to represent the band gap transition about the number of layers and Rb contents in one graph to enhance the readability.

Response: Thanks for the reviewer's valuable suggestion. We have now provided optical bandgaps of 3D $\text{Rb}_x\text{Cs}_{1-x}\text{PbBr}_3$, quasi-2D $\text{PEA}_2(\text{Rb}_x\text{Cs}_{1-x})_2\text{Pb}_3\text{Br}_{10}$ as well as $\text{PEA}_2(\text{Rb}_{0.6}\text{Cs}_{0.4})_{n-1}\text{Pb}_n\text{Br}_{3n+1}$ in **Fig. S29**. Furthermore, we now also provided the PL emission wavelengths and PLQYs of $\langle n \rangle_{\text{Rb}0.8} = 3$, $\langle n \rangle_{\text{Rb}0.6} = 3$, $\langle n \rangle_{\text{Rb}0.4} = 3$ with different content of excess RbBr in one graph (**Fig. S28**).

Fig. S29 Optical bandgaps of the perovskites with different compositions.

Action: We have now provided the optical band gaps of 3D $\text{Rb}_x\text{Cs}_{1-x}\text{PbBr}_3$, quasi-2D $\text{PEA}_2(\text{Rb}_x\text{Cs}_{1-x})_2\text{Pb}_3\text{Br}_{10}$ as well as $\text{PEA}_2(\text{Rb}_{0.6}\text{Cs}_{0.4})_{n-1}\text{Pb}_n\text{Br}_{3n+1}$ in **Supplementary Information Fig. S29**. The PL emission wavelengths and corresponding PLQYs of $\langle n \rangle_{\text{Rb}0.8} = 3$, $\langle n \rangle_{\text{Rb}0.6} = 3$ and $\langle n \rangle_{\text{Rb}0.4} = 3$ perovskites with different content of excess RbBr were provided in **Supplementary Information Fig. S28**.

REVIEWERS' COMMENTS:

Reviewer #1 (Remarks to the Author):

i believe the authors have revised the manuscript carefully, as well as addressing the comments from reviewers. I believe the current major concern is whether the major claims of novelty is sufficient for Nature Communication. As far as I understand, the primary new strategy is that the authors used a Rb-Cs alloying approach to prepare quasi-2D perovskite, and obtained a EQE 1.35% blue emission LED (454-492 nm). although the lifetime and efficiency is still low, this work represents an important progress from the spectral point of view, i.e., it advances further in shorter wavelength compared with previous literature. therefore i support the publication of this work in Nature Comm. some language should be further tightened.

Reviewer #2 (Remarks to the Author):

The authors addressed my previous comments in an appropriate way. The manuscript has been improved and can be accepted as it is in the journal of Nature Communications.

Reviewer #3 (Remarks to the Author):

The authors have addressed my comments by additional analysis and explanation. I feel that this manuscript is suitable to publish as it is.